

# Fitts' law-based identification of motor development stages for the upper limb: proof of concept in three age groups

Cristina Sanchez[1], Eloy Urendes[1], Alejandra Aceves[1], María Martínez-Olagüe[2,3] and Rafael Raya[1]

[1] Departamento de Tecnologías de la Información, Escuela Politécnica Superior, Universidad San Pablo-CEU, CEU Universities, Madrid, Spain
[2] Departamento de Fisioterapia, Facultad de Medicina, Universidad San Pablo-CEU, CEU Universities, Madrid, Spain
[3] Neuroped, Centro de Neurorehabilitación Pediátrica Integral, Madrid, Spain

## ABSTRACT

**Background:** Psychomotor development, including fine motor skills, progresses throughout childhood and stabilizes in adulthood. This process is closely tied to neurological maturation, with "reaching and pointing tasks" considered fundamental upper limb functions. According to Fitts' law, movement time (MT) depends on the task's index of difficulty (ID). From an Information Theory perspective, throughput (TP) reflects processing speed in reaching tasks, while error rate (ER) quantifies incorrect selections. As motor control improves, TP is expected to increase and ER to decrease, indicating greater efficiency and coordination. This study aimed to compare TP and ER across three age groups to assess motor control evolution.

**Methods:** Sixty participants were divided into three groups: children (5–6 years), adolescents (14–15 years), and adults (21–24 years). All participants performed a 2D reaching task on a tablet using their dominant hand, in accordance with the International Organization for Standardization (ISO) 9241-411 standard. Each participant completed 23 trials under four IDs, varying target size and distance. TP and ER were calculated and the data were statistically analyzed using an analysis of variance (ANOVA) and *post-hoc* tests to identify differences between groups.

**Results:** TP and ER showed significant differences across age groups. Children (5–6 years) had the lowest TP ($3.84 \pm 0.95$ bits/s) and the highest ER ($17.07 \pm 8.15\%$). Adolescents (14–15 years) demonstrated higher TP ($5.88 \pm 0.64$ bits/s) and lower ER ($5.06 \pm 3.13\%$), while adults (21–24 years) exhibited the highest TP ($6.46 \pm 1.05$ bits/s) and a slightly higher ER ($6.81 \pm 5.07\%$) than adolescents. A one-way ANOVA confirmed a significant effect of age on TP ($F_{2,57} = 47.18$, $p < 0.001$, $\eta_p^2 = 0.623$) and ER ($F_{2,57} = 22.1$, $p < 0.001$, $\eta_p^2 = 0.437$). *Post-hoc* comparisons revealed that children had significantly lower TP and higher ER than both adolescents and adults ($p < 0.001$). Additionally, adolescents showed significantly lower TP than adults ($p < 0.05$). However, no significant differences in ER were found between adolescents and adults.

**Conclusions:** The results indicate that TP and ER, derived from Fitts' law, effectively capture age-related differences in motor control across different developmental stages. These findings align with typical neuromotor development. Children show the lowest performance in both speed and accuracy, with adults outperforming both

Corresponding author
Cristina Sanchez,
cristina.sanchezlopezpablo@ceu.es

children and adolescents in processing speed, and adolescents demonstrating similar accuracy compared to adults. These metrics show potential for clinical and research applications, particularly in evaluating motor impairments or tracking rehabilitation progress in neurological conditions and advancing motor development research. Future studies should explore its use in clinical populations and across various age ranges to enhance assessment and intervention strategies.

## INTRODUCTION

Psychomotor development refers to the ongoing acquisition of motor skills observed in children throughout childhood, with stabilization occurring in adulthood. It involves the maturation of neurological structures, including the brain, spinal cord, nerves, and muscles. Additionally, it encompasses the learning process through which infants and children explore themselves and the world around them (*Clinica Universidad de Navarra, 2024*).

The earliest significant efforts to comprehend human psychomotor development were undertaken by *Gesell (1928, 2007)* and *McGraw (1943)*. Gesell argued that the sequence in which children acquire basic motor skills (such as reaching, crawling, and walking) follows a fixed, biologically programmed order that is resistant to environmental influences. While environmental factors may affect the rate of development, the overall sequence of milestones remains largely unaffected. Similarly, *McGraw (1943)* supported this idea by focusing on the neuromuscular maturation of infants, highlighting that motor skills develop through a natural progression shaped primarily by neurological factors. In addition, *Wade & Whiting (1986)* highlighted that motor development also involves the coordination and control of sensory, cognitive, and neurological processes, which become more refined as children grow (*Wade & Whiting, 1986*).

The term motor performance refers to a temporary state of motor behavior which is task-specific and varies based on task difficulty, environmental conditions, and the individual's physiological characteristics. However, due to the variability and complexity of fine motor tasks, establishing standardized activities for the upper limb remains challenging. Among daily activities, reaching and pointing tasks are given high priority as they represent essential movements of the upper limb. Successfully performing these tasks requires visual-motor coordination to meet speed and accuracy demands (*Robin et al., 2023*; *ISO, 2012*).

Fitts' law, introduced by *Fitts (1954)*, is a widely used predictive model in human-computer interaction (HCI), ergonomics, and kinematic studies to understand the time required to reach and point to a target, based on the ratio of distance to target size. Its effectiveness in assessing user performance across various input devices, including touch screens and styluses, which are now ubiquitous in daily life, has been extensively demonstrated (*Sanchez et al., 2021*; *Monteiro et al., 2023*; *Rowland et al., 2024*; *van Zon et al., 2020*).

The law states that movement time (MT) is linearly related to the index of difficulty (ID), which is defined by the distance to and size of the target:

$$ID = log_2\left(\frac{2A}{W}\right)$$

where $A$ represents the amplitude or distance to the target, and $W$ is the width of the target.

From an Information Theory perspective, throughput (TP) measures the speed at which information is processed during a reaching task, with higher values indicating faster processing. TP is often measured in bits per second (bits/s) and was calculated as:

$$TP = \frac{ID}{MT}$$

where MT is MT (*Plamondon & Alimi, 1997*; *Schmidt & Lee, 2020*).

Error rate (ER) quantifies the proportion of incorrect selections relative to the total number of trials, with higher values reflecting lower accuracy. It is usually computed as a percentage (%), calculated as follows:

$$ER = \frac{Number\ of\ errors}{Total\ trials} \cdot 100.$$

With TP primarily reflecting processing speed and ER capturing accuracy, these metrics are valuable for studying motor performance and developmental progression.

In fact, several studies have employed Fitts' law to study and analyze motor skills across different age groups, including its application to understand developmental aspects in younger populations and provide insights into age-related differences in movement efficiency and accuracy (*Sharif et al., 2020*; *Woodward et al., 2020a*; *Sleimen-Malkoun, Temprado & Berton, 2013*). However, fewer studies have focused on comparing TP and ER specifically in a reaching task setting across a wide range of age groups (*Carvalho et al., 2015*). If the values of these metrics fall within well-differentiated ranges based on the subject's age, they could serve as a useful tool for evaluating motor changes. This is particularly relevant in school-aged children, where developmental milestones are well-established and where perceptual development is closely linked to motor control. As children's perceptual abilities mature, they begin to more effectively process sensory information such as visual, tactile, and proprioceptive feedback. This improved processing allows them to better coordinate motor actions, resulting in more accurate and efficient movements. As a result, these perceptual changes play a critical role in enhancing motor performance over time (*Spruijt, van der Kamp & Steenbergen, 2015*). Additionally, these metrics could be used to assess motor evolution in individuals with conditions such as cerebral palsy or stroke, before and after undergoing rehabilitation (*Erkek & Çekmece, 2023*). In clinical settings, these tools are essential for quantifying impairments and monitoring functional recovery. They offer objective insights into therapeutic outcomes, thereby complementing widely used functional scales (*i.e.*, the Motor Assessment Scale, MAS) (*Ashworth-Beaumont & Nowicky, 2013*; *García-Vergara & Howard, 2014*; *Aloraini et al., 2020*).

In this context, this study built on the previous key ideas by offering a detailed comparison of TP and ER values among different age groups of healthy participants in a 2D reaching/pointing task setting. Specifically, we sought to explore the capability of TP and ER to differentiate between various stages of motor progression in healthy individuals. We hypothesize that TP and ER will capture clear age-related differences in motor control, reflecting the developmental trajectory of motor skills. Specifically, we expect children to show the lowest performance in both speed and accuracy, while adults will demonstrate significantly better performance. Adolescents, being in a relatively advanced stage of motor development, are expected to show intermediate values, with their performance likely closer to that of adults than to children.

## MATERIALS AND METHODS

### Participants

This study included 60 participants, divided into three age groups: 20 children aged 5–6 years (11 males, nine females), 20 adolescents aged 14–15 years (10 males, 10 females), and 20 adults aged 21–24 years (eight males, 12 females). These groups were selected based on the work of *Gallahue (1989)* and correspond to key motor development phases. The 5–6-year-old group represents the transition from the fundamental movement phase to the specialized movement phase, where motor skills become more refined and adaptable. Adolescents aged 14–15 years are in the specialized movement phase, particularly the application and lifelong utilization stages, where motor abilities are optimized for complex and specialized tasks. Finally, the adult group (21–24 years) represents fully stabilized motor control, serving as a reference for mature motor function. This selection ensures that the study captures motor development across distinct and well-documented milestones.

All participants used their dominant hand, which was the right one in all cases, and had no known motor or neurological impairments. The research was conducted following the ethical guidelines established in the Declaration of Helsinki, and all participants (or their legal guardians, in the case of minors) provided informed consent prior to participation. Ethical approval was obtained from the Research Ethics Committee of the San Pablo CEU University (561/21/53).

### Apparatus and task design

The experimental setup utilized a Microsoft Surface Go 3 tablet (Windows operating system; Microsoft, Redmond, WA, USA), featuring a 12.4-inch PixelSense touchscreen with a resolution of 1,920 × 1,280 pixels and a 60 Hz refresh rate. To calibrate the touchscreen, the "Settings" were accessed through the "Control Panel". Within this menu, the option to calibrate the touchscreen ("touch input") was selected. The system then prompted the user to touch specific points on the screen, ensuring that the touchscreen response was accurately mapped to the display. This process was repeated until the calibration was confirmed as accurate. This setup ensured accurate measurement of movement dynamics, making it well-suited for evaluating motor control in the context of Fitts' tasks. The experiment was conducted using the "FittsStudy" software

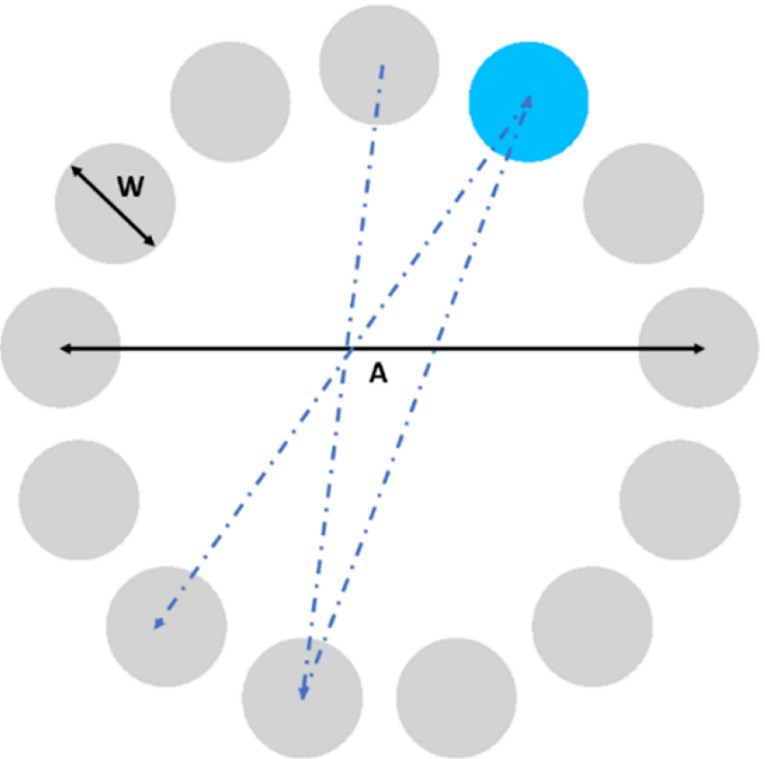

**Figure 1  ISO 9241-411 multi directional point-select test.**

(*Wobbrock et al., 2021*), which was configured to perform two-dimensional (2D) pointing tasks following the International Organization for Standardization (ISO) 9241-411 standard (*ISO, 2012*).

Participants were required to select a series of circular targets equally spaced around a circular layout using the index finger of their dominant hand. They also were explicitly instructed to complete the task "as quickly and as accurately as possible". Prior to data collection, a standardized demonstration was provided to ensure uniform task understanding. In addition, participants were encouraged to ask questions, and clarifications were given as needed to confirm their comprehension of the procedure.

Each test consisted of a randomly generated sequence of targets. Each sequence consisted of 23 trials of reaching: three were for practice (test trials), and 20 were used for data collection (effective trials). All tests started with the cursor in the center of the layout (home position), and the targets were displayed in light gray from the start of each sequence. The target to be reached was highlighted in blue as needed and if a target was missed, its color changed to red. This design choice was made to ensure visibility, even for color-blind participants (see Fig. 1). Each target was reached only once per sequence and, at the end of each sequence, participants were instructed to return to the home position.

The values of W (32 and 96 pixels) and A (256 and 512 pixels) were chosen to result in four distinct ID values (2.41, 3.41, 4.00 and 5.00) (*Carvalho et al., 2015*), giving rise to four

different conditions (tests) and providing a variety of task difficulties suitable for comparing motor control across age groups. The order in which the four conditions were chosen was randomly determined.

### Data collection and analysis

The selected metrics for comparison across age groups were the mean TP value (bits/s) and the mean ER (%). These values were obtained by averaging the metrics calculated for each participant. Specifically, each participant's TP was calculated using the average MT and ID across all their effective trials (*Zhai, 2004*). Each participant's ER was computed according to its definition, considering all errors across their effective trials.

All statistical analyses were performed using R software. First, normality tests (Shapiro-Wilk) were conducted for each group on both TP and ER to assess whether the data followed a normal distribution. Given that the sample sizes were equal across groups, a one-way analysis of variance (ANOVA) was employed to assess differences in TP and ER among the three age groups. Homoscedasticity was tested using Levene's test for both metrics. The effect size was measured using partial eta squared ($\eta_p^2$). *Post-hoc* analyses were conducted using pairwise t-tests with *p*-values corrected using the Holm-Bonferroni method to identify significant differences between specific pairs of age groups for both TP and ER. The significance level was set to 0.05 in all cases.

## RESULTS

The average values for TP and ER across the different age groups are presented as follows: for children (5–6 years), the mean TP was 3.84 ± 0.95 bits/s, and the mean ER was 17.07 ± 8.15%. In adolescents (14–15 years), the mean TP was 5.88 ± 0.64 bits/s, with a mean ER of 5.06 ± 3.13%. Finally, for adults (21–24 years), the mean TP was 6.46 ± 1.05 bits/s, and the mean ER was 6.81 ± 5.07%.

A one-way ANOVA showed a significant effect of age group on TP ($F_{2,57}$ = 47.18, $p < 0.001$, $\eta_p^2$ = 0.623) and ER ($F_{2,57}$ = 22.1, $p < 0.001$, $\eta_p^2$ = 0.437). The calculated $\eta_p^2$ values indicate large effect sizes ($\eta_p^2 > 0.14$) for both TP and ER.

*Post-hoc* comparisons indicated that children (5–6 years) had significantly lower TP compared to both adolescents (14–15 years; $p < 0.001$) and adults (21–24 years; $p < 0.001$), while adolescents had significantly lower TP than adults ($p < 0.05$). For ER, children had significantly higher values compared to both adolescents ($p < 0.001$) and adults ($p < 0.001$). No significant differences in ER were found between adolescents and adults.

To visualize these results, Fig. 2 shows the boxplot graphs of both the TP and ER values for each age group, including the statistical significance results.

## DISCUSSION

This study examines motor control development across three distinct age groups using Fitts' law-derived TP and ER. The significant differences in TP and ER between adults, adolescents, and children reflect the expected neuromotor developmental trajectories, supporting previous findings on age-related changes in motor performance, particularly in terms of processing speed and accuracy (*McGraw, 1943*; *De Meester et al., 2020*). As
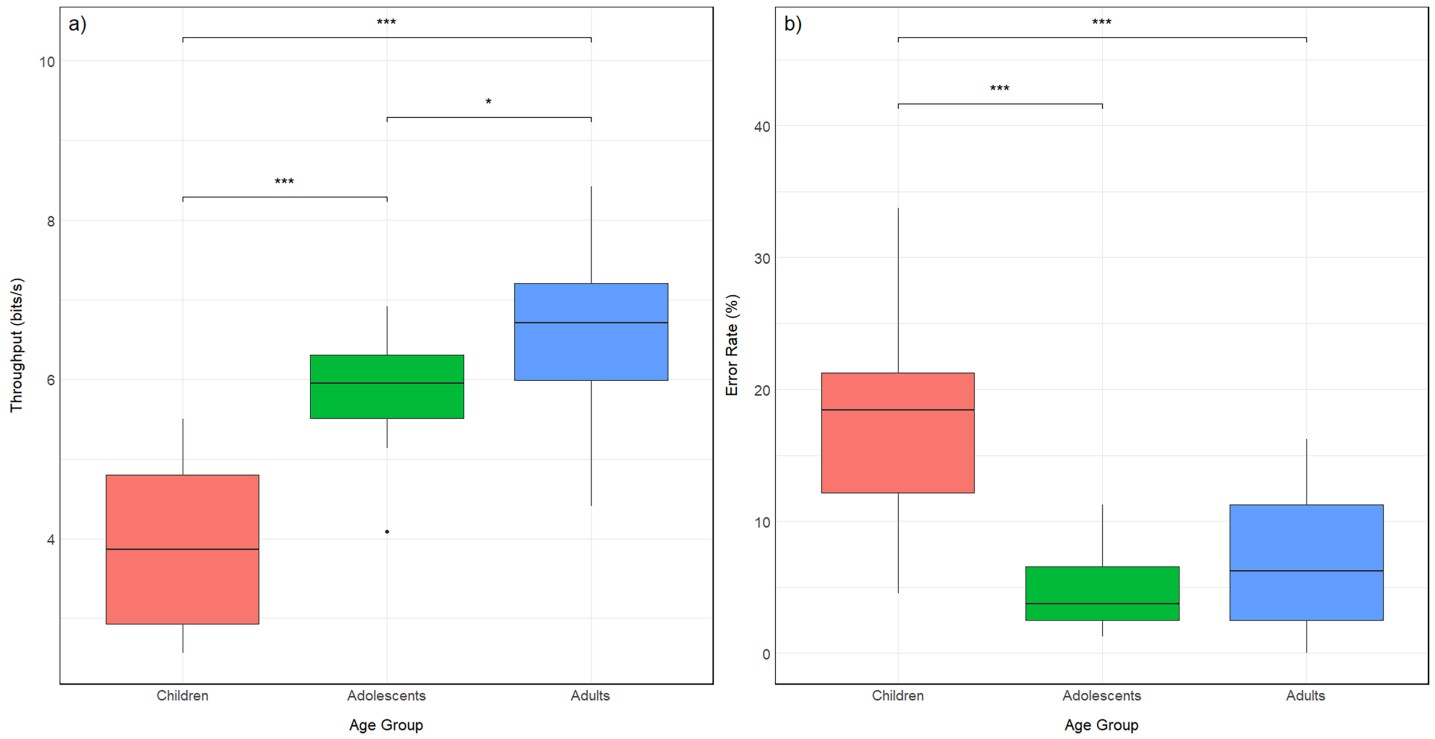

**Figure 2** **Boxplots showing the distribution of (A) throughput (bits/s) and (B) error rate (%) across three age groups: children, adolescents, and adults.** Each boxplot represents the median (central line), interquartile range (box), and whiskers extending up to 1.5 time the interquartile range. Outliers are shown as individual points. Statistical comparisons between groups are indicated with horizontal lines, where *** ($p < 0.001$) and * ($p < 0.05$) denote significant differences.               

expected, adults showed significantly higher TP values compared to adolescents and children. Additionally, both adults and adolescents had significantly lower ER values compared to children. This is in line with established research suggesting that motor control peaks in adulthood (*Gallahue, 1989*). According to Fitts' law, the trade-off between speed (related to TP) and accuracy (related to ER) defines motor performance, and adults are better able to optimize this trade-off due to fully developed neuromotor systems and more refined perceptual abilities (*DiSalvi & Kondraske, 2021*; *Parziale, Senatore & Marcelli, 2020*).

The lower TP in adolescents compared to adults supports the hypothesis that motor skills development continues through adolescence. Previous research (*Schulte et al., 2020*) suggested that the variability in motor performance during adolescence may reflect ongoing development of sensorimotor pathways, as well as less developed perceptual abilities compared to adults. Additionally, children, as expected, exhibited the lowest TP and the highest ER, showing significant differences compared to both adults and adolescents. This can be attributed to their developing motor systems, which indicate challenges in balancing speed and accuracy when integrating sensory feedback with effective motor actions. Moreover, the elevated ER in the children's group can also be explained by the hypothesis that their less developed perceptual skills may contribute to difficulty in following instructions and performing tasks accurately. Interestingly, ER did

not show significant differences between adolescents and adults. This suggests that ER, as a measure of accuracy, may be more sensitive to the developmental differences in younger individuals who are still in the early stages of refining their motor skills and may exhibit greater variability in the precision of their movements (*Wollesen et al., 2022*).

Therefore, based on the analysis conducted and supported by relevant literature, both TP and ER effectively capture key developmental differences in motor skills across the lifespan, with TP reflecting movement speed and ER indicating accuracy. These findings highlight their potential as complementary metrics for assessing motor control, with the combined analysis of both measures offering a more comprehensive assessment of motor performance (*Woodward et al., 2020b*; *Yadav et al., 2021*). Furthermore, these metrics could be valuable in both clinical and research settings for studying motor control evolution, whether for diagnosis, monitoring, or rehabilitation. Their application could contribute to a better understanding of motor development across different ages and conditions, as well as to the evaluation of interventions aimed at improving motor function.

Recent studies have demonstrated the utility of Fitts' law in clinical settings, particularly for conditions such as cerebral palsy (CP), where motor impairments are prevalent (*Gump, LeGare & Hunt, 2002*; *Borish et al., 2021*). Beyond CP, Fitts' tasks have been employed to assess motor function in various conditions, including stroke, ataxia, cognitive impairment, spinal pathologies, and obesity, highlighting their broad applicability in evaluating motor control deficits (*Malone, 2021*). TP and other derived metrics have proven sensitive to subtle motor changes, making them valuable for tracking rehabilitation progress. In our study, we observed that these metrics, particularly TP and ER, effectively captured age-related differences in motor control, reinforcing their potential for tracking motor impairments in clinical populations. Additionally, the use of digital and tablet-based tools, such as the one employed in this study, offers clinicians scalable and objective methods for assessing motor control (*Mia et al., 2024*). Moreover, for individuals with limited hand mobility who may struggle with traditional touchscreen-based tasks, alternative HCI methods could be implemented. For instance, motion-based controllers or inertial sensors (IMUs) could be used to capture upper limb movements in a more adaptable manner. Previous studies have shown that air mice and wearable IMU-based interfaces can allow individuals with motor impairments to perform reaching and pointing tasks, maintaining adherence to Fitts' law principles while providing accessible means of motor assessment (*Budhi Widodo et al., 2020*; *Brunfeldt, 2012*; *Vella & Vigouroux, 2013*).

Beyond their diagnostic and monitoring capabilities, the results of this study highlight how Fitts' tasks and derived metrics, such as TP and ER, can be instrumental in refining evaluation protocols. The significant age-related differences observed in TP and ER support the potential for these metrics to enhance rehabilitation strategies by providing more accurate and tailored assessments. Standardizing task parameters and integrating adaptive difficulty levels can enhance the precision and applicability of these measures for individual evaluations (*Kim et al., 2021*; *Ifft, Lebedev & Nicolelis, 2011*; *Aloraini et al., 2020*). Furthermore, combining Fitts' tasks with complementary assessment methods, such as kinematic analyses or machine learning approaches, could improve their predictive

value and provide deeper insights into motor function dynamics across different populations (*Donovan et al., 2022*; *Grosprêtre et al., 2023*).

Finally, based on the study's findings, it is important to acknowledge several limitations that may impact the generalizability and interpretation of the results. One key limitation is the relatively small sample size of 60 participants. However, despite this limitation, substantial differences were observed between the groups on both measures, indicating that the study had sufficient power to detect meaningful differences where they exist. Nevertheless, further research with larger sample sizes is necessary to establish normative values for the metrics analyzed. Additionally, the fact that all participants used their dominant hand may also overlook differences in performance when using the non-dominant hand, especially in populations with motor impairments. Moreover, while the study offers valuable insights into the developmental trajectory of motor control, the lack of clinical populations limits the understanding of how these metrics, particularly TP and ER, apply to patients with neurological conditions or motor disorders. Future studies could benefit from a more diverse sample, including left-handed individuals and clinical populations, to enhance external validity and clinical relevance. Larger sample sizes and a more balanced demographic representation would also help provide a more comprehensive understanding of the relationship between age, motor control, and task performance.

## CONCLUSIONS

In conclusion, this study demonstrates that Fitts' tasks and their derived metrics, specifically TP and ER, effectively capture age-related differences in motor control. Children show the lowest performance in both speed and accuracy, with adults outperforming both children and adolescents in processing speed, and adolescents demonstrating similar accuracy compared to adults. This change observed in TP and ER across age groups provides valuable insight into the developmental trajectory of motor skills and the refinement of motor control throughout the lifespan. These findings suggest that Fitts' tasks could be further explored as a potential tool for assessing motor performance and developmental patterns. Given the quantitative nature of TP and ER, these metrics hold significant clinical potential, particularly in assessing motor function and guiding rehabilitation in patients with diverse neurological conditions. Moreover, it is important to consider the role of perceptual abilities in motor performance, especially in children, where perception may influence overall motor performance.

In clinical populations, where cognitive or perceptual impairments may exist, a thorough evaluation of perception is essential to detect its impact on motor performance and improve rehabilitation outcomes. Future research should expand the application of these tasks and metrics across various clinical populations, with a focus on enhancing assessment and intervention strategies in motor disorders. In addition, further investigation could explore how both TP and ER can be incorporated into tailored rehabilitation interventions for individuals with motor impairments, and how alternative HCI methods, such as motion-based controllers or inertial sensor interfaces, could benefit patients with limited mobility.

## ACKNOWLEDGEMENTS

The authors thank the participants for their collaboration in the experiments. The authors also acknowledge the use of ChatGPT-4 (OpenAI) to enhance the writing style of the manuscript and improve the visual presentation of the figures. All intellectual contributions and critical interpretations remain the responsibility of the authors.

### Funding

This research was funded by MCIN/AEI/10.13039/501100011033/FEDER, UE, grant number PID2021-127096OB-I00. The funders had no role in study design, data collection and analysis, decision to publish, or preparation of the manuscript.

### Grant Disclosures

The following grant information was disclosed by the authors:
MCIN/AEI/10.13039/501100011033/FEDER, UE: PID2021-127096OB-I00.

### Competing Interests

The authors declare that they have no competing interests.

### Author Contributions

- Cristina Sanchez conceived and designed the experiments, analyzed the data, prepared figures and/or tables, authored or reviewed drafts of the article, and approved the final draft.
- Eloy Urendes conceived and designed the experiments, analyzed the data, prepared figures and/or tables, authored or reviewed drafts of the article, and approved the final draft.
- Alejandra Aceves conceived and designed the experiments, performed the experiments, analyzed the data, prepared figures and/or tables, authored or reviewed drafts of the article, and approved the final draft.
- María Martínez-Olagüe conceived and designed the experiments, performed the experiments, authored or reviewed drafts of the article, and approved the final draft.
- Rafael Raya conceived and designed the experiments, authored or reviewed drafts of the article, and approved the final draft.

### Human Ethics

The following information was supplied relating to ethical approvals (*i.e.*, approving body and any reference numbers):

Ethical approval was obtained from the Research Ethics Committee of the San Pablo CEU University. 561/21/53

### Data Availability

The raw data is available in the Supplemental File.

## Supplemental Information

Supplemental information for this article can be found online at http://dx.doi.org/10.7717/peerj.19433#supplemental-information.

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
