# Peer review of "Fitts’ law-based identification of motor development stages for the upper limb: proof of concept in three age groups"

_PeerJ, doi:10.7717/peerj.19433_

## Round 0.1 · original submission · Major Revisions

Dear Authors,
The reviewers have identified several critical issues that must be addressed before your manuscript can be considered for publication. These concerns primarily relate to the methodology, statistical analysis, clarity, and overall presentation.

If you choose to revise your manuscript, we kindly request that you carefully address all the feedback provided. Specifically, the results should be reanalyzed using the original Fitts’ Law Index of Difficulty (ID) equation unless a compelling justification for the current approach is presented. Additionally, the methods section should be expanded to include all necessary details, ensuring the study's reproducibility. A thorough analysis and reporting of error rates should also be conducted, with these findings integrated into the discussion to provide deeper insights into the results. Furthermore, the statistical reporting should be clarified and made more precise, with all relevant metrics presented transparently. Lastly, the manuscript would benefit from revisions to improve grammatical accuracy and eliminate ambiguities, thereby enhancing its overall readability and clarity.

In addition to the reviewers' comments, I noticed a redundancy in Figures 2 and 3. If I am correct, these figures appear to represent the same variable using two different graphical illustrations (box plots and bar plots). It is unclear why the same results are illustrated in two ways.

We look forward to reviewing a revised version that addresses these points in detail.

Kind regards,
Emiliano Brunamonti

Reviewer 1 ·

Basic reporting

The study has sufficient information in terms of introduction and is supported by a literature review. Clear and understandable language is used throughout the study. The method part of the study is appropriate for its purpose. The figures and figures used are appropriate. The results of the study seem to be consistent with the hypothesis.

Experimental design

It was mentioned that right-handed subjects were preferred in the study especially. The concept of the dominant hand, which we usually encounter, was not used here. For this reason, it is not explained why they preferred right-handed subjects in the study over dominant-hand subjects. It should be added to the method.

Could the random ordering of the four difficulty levels have caused a learning effect for the subjects who exceptionally progressed from easy to difficult? Would it be possible for them to pass the levels that become progressively more difficult in the order they are used to with fewer errors? If this happens in more than one subject, even with rondom, it may lead to misleading results for each group when comparing the test results of the groups. What would be the changes in the results if extra control groups were added to the study and all four difficulty levels were passed in the same order? Would this practice have been advantageous for younger age groups? The lack of a control group that passed the difficulty levels in the same order should be added as a limitation to the study.

Validity of the findings

Children's integration of sensory feedback into motor actions was mentioned. However, the perception process related to the test was not mentioned. Children's failure in motor control cannot be limited only to the test application. Perceptual abilities are also effective in motor control. Even if a child who cannot perceive the test protocol as well as an adult can process data, they may have problems with which data to use and where to use it. Therefore, children's perceptual level should be considered, and references in the conclusion should support the discussion.
Likewise, the mistakes made may also be related to the level of perception. The fact that the difference between adolescents and adults is lower strengthens this argument.

The study concluded that the test protocol applied in the conclusion part of the study would play a significant role in the early detection of diseases that occur later or in the early diagnosis of problems that may develop at a young age. However, this issue was mentioned very briefly in the aim section of the study, and the determination of differences between age groups was noted. In this context, the purpose of the study should be elaborated.

Additional comments

When the study is evaluated in general, it is a very good study in terms of original value and is thought to contribute to the field.

·

Basic reporting

No comment

Experimental design

No comment

Validity of the findings

No comment

·

Basic reporting

• Lines 24-25: Sentence has one or two too many clauses. Consider revision to improve clarity.
• Lines 28-30: While I understand that the abstract needs to be limited in length, I believe you should consider adding a bit more of an explanation of throughput within the abstract. With the explanations you have included, it will be difficult for a naïve reader to understand what you did based on the abstract.
• Lines 56-58: This is a run on sentence. Consider revision to improve clarity.
• Lines 62-64: I find this section to be a bit vague, and I fail to grasp the relevance of the second clause to the overall article. If you believe that this point is a necessary inclusion, I suggest expanding on what you mean in this sentence.
• Line 66: The comma on this line in not grammatically correct. Please remove.
• Line 66: Change ‘refers to’ to ‘is,’ or place a phrase like “The term ‘motor performance’…” at the start of the sentence to improve clarity
• Lines 68-70: ‘It provides a means to evaluate motor skills and can be assessed…’ is rather confusing to me. Motor performance allows for the evaluation of motor skills? And so you need to assess motor performance to assess motor skills? I’m not sure if this rhetorical construction is correct, and I’m not quite sure what it, and the preceding sentence, add to the point of the paragraph (which I interpret broadly as ‘reaching is the most critical movement for the upper limb, as it is involved in most of the activities of daily living performed with the limb’). My suggestion is to remove this paragraph and include this point briefly either as a lead in from the paragraph where you introduce motor development to the discussion of Fitts’ Law, or after the discussion of Fitts’ Law to transition to your objectives.
• Line 76: On the other hand of what? What are you contrasting Fitts’ Law to here?
• Line 86-89: I’m not sure the characterization of throughput as the ‘core metric’ derived from Fitts Law is correct, per se. There has been a lot of development in the field in the 30 years since MacKenzie wrote the chapter you reference, and a lot has been done with Fitts’ Law. Perhaps clarify that is the most important outcome for your study.
• Line 87, 92: As defined, I would disagree with your assessment that throughput is a composite measure of speed and accuracy; at least, any more than is movement time in a Fitts Task scenario. Please revise with a more precise explanation of what is important about throughput, perhaps something along the lines of ‘Throughput (TP) effectively represents the inverse of the slope of the Fitts’ Law equation for an individual target. From an Information Theory standpoint, it measures the speed at which information is processed when reaching to a target of specified amplitude and width, with a higher value representing an increase in processing speed relative to a lower value.’ This definition would also allow you to more directly discuss the relevance of throughput to it’s potential to measure motor development.
• Lines 101-106: This is a very long sentence which could be broken up, and the use of brackets reduced, to improve clarity.
• Lines 108-110: What are your overall objectives for this study (i.e., demonstrate the potential for TP to be used as a measure of motor development)? What are your hypotheses for the outcomes of this study? These have been stated in previous paragraphs, at least in a broad sense, but I find a clear statement of objectives and hypotheses helps to prepare and guide the reader through the rest of one’s work.
• Line 138: What does ‘N’ represent here? I’m a bit confused.
• Line 136: Between what?
• Lines 162-164: Please state all relevant outcomes for the ANOVA (e.g., F2,40=14.9, p<0.001, η2p = 0.427). Additionally, my recommendation is to use ‘p<0.001’ for any p-values lower than 0.001, rather than the combination of that and scientific notation, to add clarity and save space. By including the F statistic and the degrees of freedom (see my point above this one), a reader can calculate the precise p-value, if they so wish. If you feel a need to include precise p-values values, consider including them in supplemental materials.
• Line 164: By which method did you assess the effect size as ‘large?’ Not that I have an issue with that assessment, but an explanation is necessary. Additionally, I would avoid using paired mathematical symbols (i.e., ‘>>’) for emphasis.
• Line 166: This information does not contradict that which came before, so I’m not sure what ‘On the other hand’ is supposed to mean here.
• Line 167: The outcomes from your statistical tests do not directly indicate if performance is ‘better’ or ‘worse.’ That is a matter of interpretation which should be left to the discussion. These results specifically indicate that TP is higher/lower in some conditions compared to others.
• Lines 168-169: What is the direction of this difference?
• Line 170-171: The last sentence would be clearer if it read ‘These results are summarized in Figure 3.’
• Line 192: I’m not sure ‘reduced capacity’ is the most appropriate syntax here, as clearly the children fully have the capacity to perform the reaching movement you prescribed. They perform them more slowly than adults and adolescents, which is what is directly suggested by TP. If you test the error rate and find that it is significantly different for children compared to adolescents/adults (which it appears to be based on a cursory examination of Table 2), that would be evidence of a reduced capacity, as they are not adhering to your instructions to be accurate.
o More than this, if you dive into the error rates, which I believe you should, the fact that the children are both slower (which, if following instructions, would indicate an inclination towards accuracy) and less accurate (which, if following instructions, would indicate an inclination towards speed), indicates either the children were worse at following instructions (possible) and/or supports your supposition that children are indeed inherently worse at performing these tasks, as they are unable to be either as quick or as accurate as their elders.
• Line 198: ‘…particularly in clinical settings.’ Why would this be appropriate in clinical settings more than any other setting? Would it not be useful for research purposes as well? Might be worth mentioning research-oriented future directions along with the clinical ones, if you have any in mind.
• Lines 203-216: If you’re curious about other pathologies which have been tested with Fitts’ Law, you can see a brief summary of pathologies I knew to be tested with Fitts’ Law at the time I completed my Master’s thesis (Malone, 2021, ‘Fitts’ Law and Physical Disability,’ pp. 5-7).
• Your tables are quite small and may not need to exist as tables. Rather, this information should be incorporated into the text.
• Your figures look a little low detail to me. This may be an artifact of the journal creating the pdf for me to view, and thus may not reflect the true quality of your uploaded figures. However, I recommend you check to make sure your uploaded figures are high detail.

Experimental design

• Line 82: This is not a commonly used equation for Index of Difficulty in the sources I am aware of. I understand that MacKenzie (1992) argued that the version you use fits better to the data he reported. However, Plamondon and Alimi (1997), in their survey of many of the various attempts to refactor Fitts’ Law to better fit observed data, demonstrated that the original derivation for the Index of Difficulty reported by Fitts (1954; though it is an algebraic transformation of the modern form), ID=log2(2A/W), is nearly as good as it could possibly be, including for the data collected in MacKenzie’s own experiments. Additionally, Plamondon and Alimi (1997) showed that the original derivation has been shown to be very good in most situations, both for individual performances and for group means, with correlation coefficients usually above 0.9. I have not seen MacKenzie’s derivation as extensively tested for it’s fit compared to the original derivation, as I have not seen it referenced outside of MacKenzie’s own works or those of Plamondon and Alimi. The evidence which I am aware of shows: that the original derivation produces very high correlation coefficients for data derived from Fitts’ Tasks across a variety of studies, scenarios, and populations; that there is limited evidence for the MacKenzie derivation being more than marginally better than the original; and that there is far more evidence for the generalizability of the original derivation than the MacKenzie derivation (apart from Plamondon and Alimi, practically all studies published within the last 20 years I am aware of which discuss Fitts’ Law use the original derivation; see Schmidt, Lee, et al. [2019, or a more recent edition if one exists] for a modern review). Considering all this, I strongly recommend that you refactor your outcomes to use ID=log2(2A/W) and rerun your statistical tests on the refactored data, unless you have a compelling reason which supersedes the evidence stated above to use the MacKenzie Index of Difficulty derivation instead.
• Lines 116-117: Why did you choose these cohorts, specifically?
• Line 127: What was the make/model of the computer used? What were the resolution and refresh rate of the screen (these would impact how precise the measures of time and ID can be) and what was the polling rate of the touch screen? Which effector was used to reach to the targets? Was each participant given explicit instruction to ‘reach as quickly and as accurately as possible’ (or an appropriate translation)?
• Line 138: Please state all potential values for A and for W, and each of the IDs derived from these values.
o Table 1: I realize this table has some of this information in sightly different forms, but I have other issues with Table 1 (see my point about Tables for details, in the Basic Reporting section)
• Line 134: The Holm-Bonferroni method is a correction applied to the p-value of a statistical test, not a test by itself. I assume you used t-tests of some variety as your post hoc, which were corrected using the Holm-Bonferroni method. Please check if this is true and include this information in the article.
o Please also state your level of significance somewhere in your statistical methods section
• Line 144: How exactly was TP calculated by the software? How did you derive one value for each participant? Did it use a mean MT of all trials performed by each participant? Did it use the slope of the MT-ID regression? If TP for individual targets was calculated, did the TP to each of the targets significantly differ between each other? If they did, then pooling all targets together like you’ve done is inappropriate without further analysis, and more information could be gleaned about developmental stages if the targets are analyzed separately.

Validity of the findings

• You mentioned in your methods that error rates were also collected (line 147). Why mention these values if you aren’t including them in your results and why didn’t you test them? Accuracy being different between groups could have some interesting implications and are easily measured alongside throughput.
• Line 194-195: Test this statistically and report the results.
• Lines 222-225: You were not assessing the usefulness of the Fitts’ Task or TP in clinical settings, so I do not believe including these sentences is appropriate in your conclusion, at least in the way you phrased them. It is a reasonable future direction for this work, but you did not conduct a feasibility study. I would limit discussion of clinical relevance to the last sentence you have. Instead, perhaps expand on the importance of the finding that TP varies across age groups, i.e., what it shows with regards to motor development.

Additional comments

I thank the authors for their submission titled 'Fitts' law-based identification of motor development
stages for the upper limb: proof of concept in three age groups,' where they examine reaching performance in several age cohorts to provide links between performance and development. I found this an interesting and informative work, and look forward to seeing the published version. I found a few major issues, specifically 1) the use of an Index of Difficulty equation which I do not believe is nearly as well supported in the literature compared to alternatives, 2) the methods section is lacking in detail, 3) the work would benefit from a statistical analysis of the reported error rates, and 4) there are some issues with syntax and grammar, some of which I have highlighted in the Basic Reporting section. I look forward to reading the authors' responses to these and the more detailed points I have left in other sections.

Find below the sources I have cited in my review:

Plamondon, R., & Alimi, A. M. (1997). Speed / accuracy trade-offs in target-directed movements. Behavioral and Brain Sciences, 20, 279–349.

Fitts, P. M. (1954). The Information Capacity of the Human Motor System in Controlling the Amplitude of Movement. Journal of Experimental Biology, 47(6), 381–391. https://doi.org/10.1037/h0055392

Schmidt, R. A., Lee, T. D., Winstein, C. J., Wulf, G., & Zelaznik, H. N. (2019). Fitts’ Law: The Logarithmic Speed-Accuracy Trade-Off. In Motor Control and Learning: A Behavioural Emphasis (pp. 214–220). Human Kinetics.

Malone, Q. (2021). Validity and feasibility of a tri-axial accelerometer for measuring an upper limb Fitts’ Task. In MSpace. http://hdl.handle.net/1993/35781

---

## Round 0.2 · Minor Revisions

Dear Dr. Sanchez,

Your adjustments to the manuscript have been positively evaluated by the reviewers. However, Reviewer 3 suggests that further minor revisions are necessary to improve the quality of your work. Please prepare a revised version of the manuscript with the requested corrections.

Sincerely,
Emiliano Brunamonti

·

Basic reporting

No cooment

Experimental design

No comment

Validity of the findings

No comment

Additional comments

No comment.

·

Basic reporting

• In the Conclusions section of the abstract; ‘…demonstrating similar accuracy than adults.’ The word ‘than’ should be changed to just ‘to’ or ‘compared to.’
• Lines 126-127: This sentence is a little difficult for me to parse. An expansion of the idea you’re trying to present here may be warranted.
• Lines 129-132: This sentence is a run-on. Consider breaking up into smaller sentences.
• Line 134: Which hypotheses are being discussed here? You haven’t mentioned any yet, as far as I can tell.
• Line 134-142: A little quibble of mine with your tenses here. From my perspective, the ‘study,’ if it’s being discussed, occurred in the past. Presumably, you finished collecting your data and analyzing it before the reader views the article. However, if you discuss the ‘report,’ or what’s included in it, one would refer to that in the present tense, as it continues to exist and be read. Consider revision of this paragraph with these things in mind.
• Lines 136-139: This sentence has a few issues. We know that TP and ER are quantitative metrics, and this sentence appears to suggest that you are trying to figure out if they are, in fact, quantitative metrics. The clause, ‘focusing on the upper limb in this study,’ feels very out of place, and this idea should likely be placed elsewhere or otherwise integrated into the idea you’re presenting here in a different way. You also seem to suggest that your study will establish that these measures will be able to let you figure out which stage of motor development someone is in without knowing any other information about them (‘explore their ability to identify different stages…’), while you actually are trying to see if these different stages of motor development produce different TP and ER. Lastly, I would suggest changing the word ‘subject’ to the word ‘participant,’ as that is the standard syntax in use these days to refer to such individuals. My suggestion would be to cut out the middle portion of this sentence entirely and move these ideas elsewhere, leaving just something like ‘Specifically, we sought to explore the capability of TP and ER to differentiate between various phases of motor progression in healthy individuals.
• Lines 139-141: I would suggest cutting this sentence. This sounds like something which you could include in an abstract to get a reader to read the article, not something you include at the end of the introduction when you’re trying to lay out the purposes of the study.
• Lines 141-142: You have not presented any evidence to support this statement yet. I would include this in a conclusion, not at the end of an introduction.
• You’re missing any specific hypotheses. Include how you think TP and ER will change between your age ranges based on past literature and your own ideas.
• Lines 230-233: There is some redundancy in your reporting of results here. You say how children differ from adolescents and adults in the first sentence, then repeat this information in the next. Consider revision.
• Line 245: Remove ‘…as the proposed metrics,’ as I’m not sure what you mean by that.
• Lines 245-248: This should be discussed in the Introduction as a way to develop the requested hypotheses
• Line 264: remove the comma after the word ‘group.’
• Line 320: the word ‘than’ should be changed to ‘as’ or ‘compared to’ or something similar.
• Line 321: Since you didn’t examine variability, it’s a bit confusing to use the word ‘variation’ here. I would suggest a word like ‘change’ or ‘difference.’
• Lines 330-337: In my mind, a conclusion should serve as a way to highlight the most important outcomes of your study to leave the reader (and especially someone who’s skimming) remembering the main take-away points. So, suggestions for future directions do not belong here. I would fold this paragraph into the discussion, instead.

Figures
• The text on your plots is too small to be read comfortably. Please increase the text size to make the text legible while viewing the figure as a whole.
• I would suggest having children on the left side and the adults on the right. This would present the information as a ‘timeline of development’ of sorts, which makes more sense to me.
• The two results figures (2& 3) are redundant, as the present the same information. My suggestion is to move the statistical information to Figure 2 and get rid of Figure 3, as Figure 2 has more information.

Experimental design

• Please include the sex/gender ratios of each examined group.
• The task details are a little lacking. Did participants return to touch the home position before the start of every trial? Were the potential target locations marked in some way at all times? Just during a trial? Never, unless a target was the objective for a trial (note that denoting target locations while they are not the objective target can change task outcomes compared to a ‘normal’ Fitts’ Task; see Adam et al., 2006 or Malone et al., 2023 if you’re curious)?
• How many times was each target reached to?

Validity of the findings

• Please add in a discussion of why you believe that error rate did not vary between adults and adolescents, while TP did. I feel this is an interesting finding which deserves to be discussed further.
• Lines 277-299: This is all great, but please tie this information back to the results of your study. How do your results facilitate these potential approaches?
• Lines 302-304: While I agree that this data should not be considered ‘normative,’ you did find substantial differences on your two measures between groups, suggesting that you are sufficiently powered and are detecting differences where they exist. I would suggest rephrasing this away from statistical problems, which I do not believe you have, and more towards making sure the reader does not mistake the presented data as normative.

Additional comments

I thank their authors for their excellent revisions. I believe the article is in a much better place now, and my comments on this round mostly have to do with issues of clarity. The only ‘major’ issues remaining are the lack of clear hypotheses and that the methods regarding the task procedure need to be expanded to help with reproducibility. I look forward to seeing (what I would assume to be) the nearly final version!

References
Adam, J. J., Mol, R., Pratt, J., & Fischer, M. H. (2006). Moving farther but faster—An exception to Fitts’s law. Psychological Science, 17(9), 794–798. https://doi.org/10.1111/j.1467-9280.2006.01784.x
Malone, Q., McNeil, C. J., Passmore, S. R., Glazebrook, C. M., & Dalton, B. H. (2023). A violation of Fitts’ Law occurs when a target range is presented before and during movement. Experimental Brain Research, 241(10), 2451–2461. https://doi.org/10.1007/s00221-023-06687-6

---

## Round 0.3 · accepted · Accept

I believe that the current revision has addressed all the latest points raised by the reviewers.